# Primary Prevention Strategy for Non-Communicable Diseases (NCDs) and Their Risk Factors: The Role of Intestinal Microbiota

**DOI:** 10.3390/biomedicines12112529

**Published:** 2024-11-05

**Authors:** Itzel Ivonn López-Tenorio, Óscar Rodrigo Aguilar-Villegas, Yoshua Espinoza-Palacios, Lorena Segura-Real, Berenice Peña-Aparicio, Amedeo Amedei, María Magdalena Aguirre-García

**Affiliations:** 1Unidad de Investigación UNAM-INC, División de Investigación, Facultad de Medicina UNAM, Instituto Nacional de Cardiología Ignacio Cháve, Mexico City 14080, Mexico; itzeltenorio20@hotmail.com (I.I.L.-T.); oscaraguilar635@gmail.com (Ó.R.A.-V.); yoshua9509@gmail.com (Y.E.-P.); lorenafacmed@gmail.com (L.S.-R.); 2Consulta Externa Endocrinología, Instituto Nacional de Cardiología Ignacio Chávez, Mexico City 14080, Mexico; pa.berenice@gmail.com; 3Department of Experimental and Clinical Medicine, University of Florence, 50134 Florence, Italy; amedeo.amedei@unifi.it; 4Network of Immunity in Infection, Malignancy and Autoimmunity (NIIMA), Universal Scientific Education and Research Network (USERN), 50134 Florence, Italy

**Keywords:** intestinal microbiota, non-communicable diseases, hypertension, type 2 diabetes, obesity, dyslipidemia

## Abstract

Non-communicable diseases (NCDs) are the leading cause of morbidity and mortality worldwide. These conditions have numerous health consequences and significantly impact patients’ lifestyles. Effective long-term treatment is essential since NCDs are irreversible. Therefore, primary healthcare must be both exclusive and of the highest quality, ensuring comprehensive care. The primary goal should be to improve quality of life with a focus on patients, families, and communities, as most of these diseases can be prevented and controlled, although not cured. Several factors have been linked to individual health, including social, cultural, and economic aspects, lifestyle, and certain environmental factors, including work, that can have positive or negative effects. More of these variables may contribute to the onset of NCDs, which are defined by their chronic nature, propensity for prolongation, and generally slow rate of progression. Examples of NCDs include hypertension, type 2 diabetes (T2D), dyslipidemia, and fatty liver disease linked to metabolic dysfunction. The onset of these diseases has been associated with an imbalance in certain microbial niches, such as the gut, which hosts billions of microorganisms performing multiple metabolic functions, such as the production of metabolites like bile acids (BAs), short-chain fatty acids (SCFAs), and trimethylamine N-oxide (TMAO). Therefore, lifestyle changes and personal habits can significantly impact the gut microbiota (GM), potentially preventing chronic diseases associated with metabolism. NCDs are highly prevalent worldwide, prompting increased attention to strategies for modifying the intestinal microbiota (IM). Approaches such as probiotics, prebiotics, synbiotics, and fecal transplantation (FMT) have demonstrated improvements in the quality of life for individuals with these conditions. Additionally, lifestyle changes and the adoption of healthy habits can significantly impact IM and may help prevent chronic diseases related to metabolism. Therefore, the main aim of this review is to analyze and understand the importance of microbiota intervention in the prevention of non-communicable diseases. R3:A1

## 1. Introduction

Primary healthcare (PHC) emerges as an effective strategy to increase problem-solving capacity and provide preventive information by shifting from a disease-based design to a population-centered approach. This allows for the identification of individuals at risk of developing non-communicable diseases (NCDs) [1].

NCDs have a high mortality rate, registering 41 million deaths per year, that is, 74% of deaths worldwide. These diseases are characterized by remaining for a long period of time and evolving slowly [2]. These pathologies significantly impact the quality of life of patients and require long-term treatment and care [3]. NCDs result from a combination of genetic, physiological, and environmental factors. Key risk factors for the development of NCDs include systemic arterial hypertension (SAH), dyslipidemia, hyperglycemia, overweight, and obesity. However, due to the failure of primary prevention strategies to control these risk factors, new dangerous elements implicated in the NCDs development [2] have been sought, including alterations in gut microbiota (GM), which plays a critical role in this context since its involvement in various functions within the body [4].

The intestinal microorganisms play an essential role in the host immune modulation by participating in various functions, such as inhibiting pathogen colonization, maintaining the intestinal barrier, and producing metabolites [5,6]. Additionally, the GM is involved in nutrient assimilation, the degradation of indigestible plant compounds, and energy balance [5,7].

GM imbalances (dysbiosis) have been associated with non-communicable metabolic diseases, such as hypertension, type 2 diabetes (T2D), dyslipidemia, and metabolic dysfunction-associated fatty liver disease, all of which are components of metabolic syndrome. These diseases, along with cardiovascular diseases such as coronary artery disease, stroke, and peripheral artery disease, form an interconnected group of metabolic and cardiovascular disorders that represent the leading causes of morbidity and mortality worldwide [8].

Therefore, it is crucial to implement preventive interventions targeting modifiable risk factors to address this set of diseases’ effectively. Lifestyle interventions can influence the diversity and richness of the GM, playing a role in primary prevention to improve the NCDs management. This review focuses on the GM role in metabolic health, its impact on the pathophysiology of non-communicable metabolic diseases, and presents some lifestyle-oriented strategies and their effects on the GM and metabolic health.

## 2. Role of the Gut Microbiota and Its Metabolites in Health and Metabolic Disease

The gut microbiota constitutes a complex ecosystem, primarily composed of bacterial families such as Firmicutes, Actinobacteria, Bacteroidetes, and Proteobacteria, among others [9], whose relative proportions vary according to the host’s geographical location [9]. The term “eubiosis” describes the balance of the microbial ecosystem, characterized by a predominance of beneficial bacteria, while “intestinal dysbiosis” implies a disruption in this balance, with a decrease in beneficial bacteria and an increase in those associated with diseases, such as the *Clostridiaceae* and *Enterobacteriaceae* groups [10,11].

The GM has the metabolic capacity to process dietary components and other substances partially digested by the host [12], participating in a wide variety of functions to metabolize and synthesize products that can positively or negatively impact human health [12,13,14]. It has been demonstrated that the microbiota influences human metabolic pathways through the production of metabolites and that dysbiosis can affect their production. Some of the metabolites of interest include bile acids (BAs), short-chain fatty acids (SCFAs), trimethylamine N-oxide (TMAO), and choline derivatives, among others [15,16] (Figure 1).

### 2.1. Bile Acids (BAs)

Bile acids originate from cholesterol and are classified into two categories: primary and secondary [17]. Primary BAs are synthesized in the liver and excreted into the small intestine to facilitate fat emulsification, while secondary BAs are generated through the deconjugation of primary BAs by gut microbiota bacteria, primarily associated with the *Clostridium* genus [17,18]. BA transporters reabsorb approximately 95% of BAs for reuse in the liver, while the less soluble secondary BAs are eliminated [17].

BA synthesis constitutes an important pathway for cholesterol elimination and is regulated by farnesoid X receptor (FXR) signaling [19]. One enzyme whose activity is affected by dysbiosis is bile salt hydrolase. The decrease in its activity caused by dysbiosis results in reduced conversion of primary BAs to secondary BAs, leading to an increase in the amount of reabsorbed BAs in circulation. This increase in BAs activates the hepatic FXR, thereby inhibiting the hepatic LXR receptor that plays a crucial role in promoting the conversion of cholesterol into BAs, activating the reverse cholesterol transport mechanism in enterocytes and hepatocytes, and favoring its accumulation. Additionally, LXR contributes to glycemic control by decreasing gluconeogenesis [19].

### 2.2. Short-Chain Fatty Acids

GM metabolizes complex dietary carbohydrates through fermentation, producing SCFAs, such as acetate, propionate, and butyrate [20]. These SCFAs are primarily produced by bacterial families like *Firmicutes*, *Bacteroidetes*, and other anaerobes [21]. SCFAs play a crucial role in modulating host metabolic pathways, exhibiting anti-inflammatory effects and serving as an energy source for intestinal epithelial cells. [22,23] For example, butyrate activates hypoxia-inducible factor 1 (HIF-1), promoting gene expression like erythropoietin (EPO), improving intestinal barrier integrity, and inhibiting nuclear factor kappa B (NF-kB) activity, thus reducing pro-inflammatory cytokine production [24].

Recent studies suggest that SCFA production varies with the host’s metabolic state. In patients with obesity or type 2 diabetes, reduced SCFAs have been linked to increased low-grade inflammation associated with these metabolic states. However, other studies have found elevated SCFA levels in the intestines of obese patients, suggesting a paradoxical effect where SCFAs may contribute both to metabolic regulation and to energy accumulation [25]. Hence, SCFAs play a dual role: while their production benefits inflammatory balance under normal conditions, excessive accumulation may be involved in obesity and insulin resistance.

### 2.3. Trimethylamine N-Oxide (TMAO)

In recent years, trimethylamine N-oxide (TMAO) has emerged as a metabolite of great interest due to its association with inflammatory diseases, especially cardiovascular diseases.

TMAO is formed in the liver from trimethylamine (TMA), which is produced by the gut microbiota (GM) from phosphatidylcholine and L-carnitine found in dietary foods such as red meat, cheese, and eggs [25]. The enzyme flavin-containing monooxygenase 3 (FMO3) catalyzes the conversion of TMA to TMAO. This TMA/FMO3/TMAO axis has been associated with increased insulin resistance and hepatic lipogenesis [25,26]. The TMAO levels in blood are elevated in consequence of gut dysbiosis, since the members of the *Firmicutes*, *Actinobacteria,* and *Proteobacteria* phyla, that main producer of TMA precursor, are characteristic of gut dysbiosis.

TMAO can trigger inflammation pathways, increasing the expression of IL-1β and IL-6 through the activation of NF-κB. This results in a systemic pro-inflammatory state that can trigger or worsen diseases where inflammation plays a significant role, such as atherosclerosis, diabetes, and hypertension.

For example, elevated blood TMAO levels have been observed in some patients with acute myocardial infarction [27]. Specifically, ST-elevated myocardial infarction (STEMI) is characterized by elevated levels of IL-6, IL-1β, and C-reactive protein. Elevated levels of TMAO are considered a risk factor for STEMI.

In diabetes, it has been found that higher TMAO levels are in diabetes patients, in comparison with prediabetes and non-diabetes patients. It has been reported that deletion of FMO3, an enzyme responsible for TMA to TMAO conversion, increases glucose levels and insulin resistance [27,28].

### 2.4. Lipopolysaccharide

Lipopolysaccharide (LPS) is a component of the outer membrane of Gram-negative bacteria and triggers the activation of an inflammatory response through recognition by toll-like receptor 4 (TLR4) [29]. The gut microbiota is largely composed of Gram-negative bacteria, which are a significant LPS source. Various studies show that high-fat diets can cause intestinal dysbiosis, which has been associated with an increase in Gram-negative bacteria and consequently an increase in LPS levels [29,30]. This phenomenon is also observed in patients with metabolic diseases such as diabetes mellitus and could be attributed to the alteration in intestinal permeability associated with these conditions [31].

## 3. Systemic Inflammation in Metabolic Diseases

Physiologically, depending on the stimulus, inflammation serves to protect the host from infections, repair damaged tissues, or adapt to stressors to restore homeostasis. However, prolonged stimulation can lead to pathological states ranging from tissue damage, fibrosis, and loss of tissue function to autoimmunity, the development of neoplasms, and NCDs [32].

Most metabolic diseases share a common inflammatory basis responsible for the damage experienced by the organism as well as the associated complications [33]. Systemic inflammation is detected by an increase in leukocytes, C-reactive protein, and pro-inflammatory cytokines such as IL-1, IL-6, and TNFα, which are used clinically due to their relevance in estimating the complications’ risk. This underscores the fundamental role of inflammation in the development and progression of metabolic diseases such as T2D, obesity, and hypertension, among others [34].

Metainflammation is a relevant event of chronic metabolic diseases, such as obesity and diabetes, that goes beyond systemic inflammation. It is characterized by chronic low-grade inflammation triggered by metabolic imbalance, leading to a reprogramming of immune cells, affecting macrophages in particular [35]. These cells, which normally help maintain tissue homeostasis, undergo a shift from an anti-inflammatory (M2) to a pro-inflammatory (M1) phenotype during metainflammation. This change exacerbates insulin resistance and other metabolic disorders.

Macrophages play a crucial role in this process by altering their metabolic programming. In obesity and type 2 diabetes, they become dependent on glycolysis, which promotes the production of pro-inflammatory cytokines and contributes to insulin resistance and tissue damage. On the other hand, M2 macrophages, which depend on oxidative phosphorylation (OXPHOS) and fatty acid oxidation (FAO), are associated with anti-inflammatory effects. Reprogramming these macrophages back to an anti-inflammatory state is a key goal in the management of metabolic diseases [36].

Dietary and lifestyle interventions have been shown to influence this reprogramming. For example, calorie restriction, exercise, and anti-inflammatory diets can promote the shift of macrophages to the M2 phenotype. This reduces systemic inflammation and improves metabolic outcomes such as insulin sensitivity and lipid metabolism [35,36].

In addition to what has already been mentioned, the metabolites present in each person depend on the GM eubiotic or dysbiotic function, which can have positive or negative repercussions on the inflammatory state. In individuals with eubiosis, higher amounts of metabolites such as SCFAs and BA are observed, which are important for preventing uncontrolled inflammatory responses. Conversely, in subjects with dysbiosis, certain bacterial genera can increase levels of metabolites such as TMAO or LPS, which can activate the inflammatory response and trigger cytokines’ production, thereby contributing to the pathological effects observed in metabolic diseases [37].

## 4. GM Influence on Risk Factors for Non-Communicable Metabolic Diseases

As previously reported, non-communicable metabolic diseases, characterized by imbalances in key enzymatic systems of intermediary metabolism, often arise due to disorders in endocrine or metabolically active organs [17]. Obesity, dyslipidemia, and hypertension are closely linked to metabolic syndrome, which increases the likelihood of developing diseases including T2D, metabolic dysfunction-associated fatty liver disease, and cardiovascular diseases, major causes of mortality worldwide [38]. Some GM metabolites can influence the human inflammatory profile and metabolism and have an important role in the development of these metabolic diseases (Figure 1).

### 4.1. Metabolic Syndrome

Metabolic syndrome (MS) is characterized by a combination of metabolic risk factors, including abdominal obesity, dyslipidemia, insulin resistance, and hypertension, considered a pathophysiological entity rather than just the sum of individual diseases [39]. Its prevalence has increased globally in recent years, affecting up to 20–40% of the adult population in various countries [40].

Nutritional treatments play a crucial role in MS management, emphasizing the relevance of a diet rich in fruits, vegetables, fish, nuts, legumes, and whole grains, along with an adequate intake of proteins and monounsaturated fats. Sodium restriction and a low-carbohydrate diet with a reduced glycemic load have also been associated with improvements in MS [41].

In addition to these mentioned risk factors, MS patients with MS exhibit GM alterations, which can trigger metabolic imbalances such as low-grade inflammation induced by intestinal dysbiosis. This condition weakens the intestinal barrier and promotes insulin resistance, thus contributing to the MS progression [42].

Excessive sugar consumption negatively impacts the gut microbiome, triggering more adverse effects on the host’s intestinal health, including slower intestinal transit, microbial composition imbalances, and increased endotoxin production. Additionally, foods high in added sugars, being empty calories, displace more nutritious foods, contributing to both overeating and malnutrition [5]. Intestinal dysbiosis disrupts the intestinal mucosa, crucial for barrier function, resulting in increased intestinal permeability. This alteration in the bacterial balance and immune system promotes the translocation of bacterial fragments, thus promoting metabolic endotoxemia, low-grade systemic inflammation, and insulin resistance [43,44].

In the long term, these alterations can trigger health problems such as weight gain, type 2 diabetes, obesity, metabolic syndrome, metabolic dysfunction-associated fatty liver disease (MAFLD), irritable bowel syndrome, and cardiovascular diseases [45].

### 4.2. Obesity

Obesity, a global health issue, is influenced by the metabolism of certain bacteria that facilitate the extraction of energy from food, increasing the storage of fat deposits in adipose tissue and contributing to the rise in body mass index (BMI). This ability to efficiently extract energy represents a selective advantage that favors a thrifty phenotype, especially in environments where food resources are scarce. However, this adaptation contributes to energy imbalance and low-grade chronic inflammation associated with obesity [46].

Changes in GM diversity and abundance can alter cellular metabolism. For example, increased LPS levels have been observed in the bloodstream of overweight and obese individuals, suggesting a potential malfunction of the intestinal barrier. This phenomenon also affects adipose tissue and insulin resistance [47].

The GM plays a fundamental role in obesity development, as demonstrated by studies in animal models. For instance, in an experiment where wild and germ-free mice were fed the same high-fat diet, the wild mice developed obesity while the germ-free mice did not, suggesting that the absence of bacteria allowed them to resist diet-induced obesity [48].

A higher *Firmicutes/Bacteroidetes* ratio has been observed among obese subjects, suggesting an association between GM composition and body weight status. This relationship is influenced by a typical Western diet, characterized by high levels of sugar and fats, which favors the growth of these organisms [49,50].

In addition, the dysbiosis observed in obesity results in the disruption of the intestinal barrier and increased intestinal permeability, leading to low-grade inflammation. This inflammation is a characteristic associated with the pathology, where an increase in *Firmicutes* and a decrease in *Bacteroidetes* have also been reported [51].

Furthermore, obesity is positively associated with the genus Clostridium and species such as *Eubacterium rectale*, *Clostridium coccoides*, *Lactobacillus reuteri*, *Clostridium histolyticum*, and *Staphylococcus aureus* [52], while the administration of *Akkermansia muciniphila* prevents the development of obesity and its complications [53].

Reports mention that certain bacterial genera are enriched or decreased in the intestines of overweight or obese subjects [54,55]. Specifically, a differential proportion has been observed among the genera *Veillonella*, *Bulleidia*, and *Oribacterium*, which increase in a pathological condition [56,57].

### 4.3. Dyslipidemias

Dyslipidemia is a disease characterized by an increase in the concentration of total cholesterol, LDL cholesterol, or triglycerides, or a decrease in the concentration of HDL cholesterol in plasma [58]. This condition increases the risk of atherosclerotic cardiovascular disease. Given its multifactorial development, a comprehensive approach to treatment is essential [59].

Comparative animal studies with germ-free and conventional mice have demonstrated that the first exhibit less hypercholesterolemia and excrete more cholesterol in the liver and feces when fed a high-fat, high-carbohydrate Western diet. Additionally, LPS exposure significantly increases serum triglycerides in mice [41]. Lipid metabolism is regulated by GM through various mechanisms: one is the SCFA production that regulates the host’s immune and energy homeostasis through binding to G-proteins. These SCFAs regulate the synthesis and oxidation of fatty acids in tissues via the activation of peroxisome proliferator-activated receptors (PPAR). Additionally, other microbial metabolites have been directly linked to dyslipidemia, including TMAO, conjugated linoleic acid (CLA), and secondary bile acids [42].

In patients with dyslipidemias, a lower bacterial diversity has been found, along with an increase in the relative abundance of LPS-producing genera such as *Megasphaera* and *Escherichia-Shigella*, and a decrease in SCFA-producing genera such as *Christensenellaceae R-7*, *Ruminococcaceae UCG-014*, *Akkermansia*, and *Eubacterium eligens* [43].

This strongly supports the idea that GM modulation could be a synergistic therapeutic strategy for treating dyslipidemia [44].

### 4.4. Systemic Arterial Hypertension

Systemic arterial hypertension (SAH) is a highly prevalent disease worldwide and is considered the most important risk factor for all causes of morbidity and mortality. The SAH etiology is multifactorial and closely related to metabolic dysfunction [45,46].

It has been proposed that GM has the ability to modify blood pressure through its metabolites. When fiber-containing products are ingested, the microbiota produces SCFAs and secondary BAs. These metabolites lead to an increase in regulatory T cells and a decrease in T helper (Th)17 cells, which results in a decrease in cytokines such as IFNγ, IL-1β, and IL-17, which reduces inflammation and arterial fibrosis. Additionally, SCFAs can inhibit the renin–angiotensin–aldosterone system and decrease sympathetic nervous system activity via the vagus nerve [46,47].

Another mechanism related to immunity is mediated by TMAO; in addition, microbiota-derived uremic toxins, such as indoxyl sulfate and p-cresyl sulfate, are associated with the development of oxidative stress and endothelial dysfunction. This leads to pathological changes in the arterial walls, causing an increase in vascular resistance, elevated blood pressure, and cardiovascular complications. Additionally, TMAO alone can prolong the hypertensive effect of angiotensin II [46].

In addition, TMAO induces inflammation, damaging endothelial cells. This damage leads to the activation of the NLRP3 inflammasome and the production of inflammatory cytokines IL-1β and IL-18, which inhibit nitric oxide synthase, increasing systemic blood pressure [48].

In germ-free animal models, when compared to germ-free mice with microbiota acquisition, relative hypotension and a marked reduction in vascular contractility are observed, evidencing the regulatory GM role in blood pressure [49].

It has been observed that in SAH patients, there are alterations in GM composition. In these patients, there is an increase in *Firmicutes* and a decrease in *Bacteroidetes*, as well as a higher abundance of *Lactobacillus* and *Akkermansia*. In contrast, a decrease in *Roseburia* and *Faecalibacterium*, as well as butyrate-producing commensals such as *Lachnospiraceae* and *Ruminococcaceae*, is observed [50,51]. Another study found that an increase in *Bacteroidetes* might be associated with a decrease in SCFA-producing genera such as *Faecalibacterium*, *Blautia*, and *Anaerostipes* and an increase in those that can produce LPS like *Megamonas* [52].

In a study conducted on obese SAH patients, two bacterial niches, the oral and intestinal, were analyzed. It was observed that diversity was different among these patients, highlighting that in the oral cavity, the *Kluyvera* genus predominated, while in the intestinal niche, the *Firmicutes* genus showed greater abundance [53].

### 4.5. Type 2 Diabetes

T2D is a chronic metabolic disorder caused by insufficient insulin production or the inability to properly metabolize glucose despite adequate insulin production, leading to elevated blood glucose levels. This condition is characterized by low levels of insulin receptors and insulin resistance. It has become one of the major public health problems worldwide [54]. Despite the preventive efforts implemented by various health programs, the prevalence of the disease and its severe complications, as well as the associated mortality, continue to rise [55].

GM dysbiosis plays a role in the T2D pathogenesis, as demonstrated by multiple studies showing the relationship between this disease and GM function or composition changes [56].

Cross-sectional studies in humans find differences in the GM composition of patients with T2D or prediabetes compared to healthy subjects. The genera *Bifidobacterium*, *Bacteroides*, *Faecalibacterium*, *Akkermansia*, and *Roseburia* show a negative association with type 2 diabetes, while the genera *Ruminococcus*, *Fusobacterium*, and *Blautia* are positively associated [57]. Additionally, the relationship between intestinal dysbiosis and the T2D progression has been demonstrated through animal models of fecal microbiota transplantation [58].

Some potential mechanisms of the effects of the microbiota involved in T2D are related to LPS. Its effects are attributed to a certain degree of systemic inflammation after the recognition of TLRs, leading to subsequent cytokine production. It has been observed that this results in negative effects on insulin sensitivity and glucose homeostasis [59].

The microbiota not only plays a role in T2D but has also been described in relation to its complications [58]. Studies have been conducted on its involvement in diabetic nephropathy, diabetic retinopathy, and diabetic neuropathy [60].

Interventions based on microbiota composition, such as fecal transplantation from glucose-intolerant mice, induce glucose intolerance in healthy mice, supporting the causal GM role in T2D. In a study of human fecal microbiota transplantation (FMT), the transfer of fecal material from lean donors to individuals with metabolic syndrome resulted in an increase in intestinal microbial diversity and improved insulin sensitivity. These findings support the hypothesis that GM dysbiosis contributes to metabolic dysfunctions [61].

### 4.6. Metabolic Dysfunction-Associated Fatty Liver Disease

Metabolic dysfunction-associated fatty liver disease (MAFLD) is strongly related to MS, T2D, and obesity. This condition is due to dysfunction in adipose tissues, characterized by the migration of immune cells (macrophages and CD4+, CD8+ T cells, dendritic cells, and natural killer cells), increased expression of inflammatory signaling receptors (TLR4 and NLPR3), and altered insulin signaling pathways in the liver, muscle, and adipose tissue [61,62].

As previously mentioned, GM plays an essential role in the development of various diseases, and MAFLD is no exception. There is evidence from animal models demonstrating the causal GM role in the MAFLD development. In one study, fecal transplantation was performed on obese mice with or without steatosis. The results showed that healthy mice, when transplanted with the fecal matter of diseased mice, developed hepatic steatosis and inflammation [62,63].

The microbiota could contribute to the MAFLD development by increasing intestinal permeability, leading to the translocation of the bacterial membrane, increased LPS, and the action of microbial metabolites [64,65].

Additionally, the microbiota in MAFLD patients has been characterized, and microbial signatures have been found that relate not only to the presence of this disease but also to its severity, highlighting an increase in the abundance of *Bacteroides* and *Ruminococcus* and a decrease in *Prevotella*, as well as a decrease in alpha diversity [66,67].

### 4.7. Cardiovascular Diseases

Cardiovascular diseases, especially ischemic heart disease and cerebrovascular disease, are the leading causes of death worldwide [68]. Their high prevalence and elevated mortality rates make them a significant public health problem [69]. Studies have revealed the relationship between GM changes and their impact on the development of these diseases [70].

TMAO is the most studied metabolite due to its close relationship with cardiovascular diseases [71]. It has been observed that TMAO is mainly implicated in atherosclerosis, where it alters the metabolism of bile acids and cholesterol, leading to their accumulation. Additionally, TMAO promotes the formation of foam cells and increases platelet reactivity [72,73]. These mechanisms contribute to the formation of atheroma plaques that can obstruct coronary and cerebral arteries.

In studies with mice with heart failure fed diets containing TMAO, an increase in heart growth, pulmonary edema, fibrosis, and a decrease in ejection fraction were observed compared to the control group [74]. Furthermore, elevated TMAO levels have been associated with an increased cardiovascular risk, being found to be increased in patients who have suffered an acute myocardial infarction [75,76].

The GM composition in healthy individuals and those with heart diseases has been the topic of recent studies. It has been observed that patients with cardiovascular diseases show a different microbiota, characterized by an increase in the genera Proteobacteria and *Pseudomonadaceae*. In patients with acute myocardial infarction, the presence of bacteria in the blood, predominantly of the genera *Streptococcus*, *Bacteroides*, *and Lactobacillus*, has been detected [77].

Additionally, a correlation has been found between higher concentrations of translocation products such as LPS and D-lactate, with increased inflammation and a rise in cardiovascular events. For example, elevated TMAO levels have been associated with the genera *Prevotella* and *Bacteroides*, according to reports by Koeth and colleagues [78].

These findings underscore the diversity of functions that different GM species can play in the development of cardiovascular diseases and in adverse events following the disease onset. It is evident that GM plays a crucial role in cardiovascular health and could represent a promising area for future research, diagnosis, and treatment.

## 5. Lifestyle Interventions and Their Impact on Gut Microbiota

The GM composition is determined by multiple factors, ranging from microRNAs and the production of immunoglobulin A (IgA) to antimicrobial peptides secreted by intestinal epithelial cells. Additionally, factors such as age, diet, and lifestyle play a crucial role in shaping the microbial composition of the gut. In this regard, lifestyle changes and healthy habits can have a significant impact on the GM, which in turn can help prevent and influence the development and management of chronic diseases related to metabolic dysfunction [77].

### 5.1. Dietary Interventions

A crucial element influencing the microbiota is the diet. For this reason, a diet aimed at enriching beneficial bacteria is essential in a preventive strategy, as there is overlap between dietary factors related to chronic diseases and their impact on the GM [78]. In detail, consuming a diet rich in dietary fiber, through the intake of whole plant foods such as vegetables, whole grains, fruits, legumes, and nuts, provides fibers that serve as a substrate for beneficial bacteria. This can promote intestinal bacterial diversity and richness, as well as an increase in genera that produce SCFAs and improve glucose and lipid parameters in patients with metabolic diseases [79,80].

Another relevant dietary factor is polyphenolic compounds, including flavonoids, tannins, and phenolic acids. These compounds are present in a variety of foods such as cereals, coffee, fruits, medicinal plants, microalgae, green tea, blackberries, blueberries, strawberries, vegetables, spices, and nuts. They have been shown to have benefits related to metabolic diseases, as well as promoting the growth of beneficial intestinal bacteria like *Lactobacillus* and *Bifidobacterium* and inhibiting pathogenic species [81,82].

On the other hand, omega-3 polyunsaturated fatty acids, essential in the human diet and present in vegetable oils, seeds, fish, and seafood, have shown health benefits, including a positive GM modulation. These fatty acids reduce the growth of enterobacteria, increase Bifidobacterium and SCFA production, and inhibit the inflammatory response associated with metabolic endotoxemia [83,84].

### 5.2. Exercise

Physical exercise is known for its numerous health benefits, including reducing the risk of cardiovascular diseases, improving mental health, and strengthening the musculoskeletal system. In recent years, its relationship with gut health has been added to its benefits, showing that proper exercise leads to decreased intestinal permeability, increased bacterial diversity, and higher SCFA production. It has been observed that the microbiota of physically fit individuals shows increased abundance of butyrate-producing bacteria belonging to the genus *Firmicutes* [85,86]. Another study, conducted on women, documented an increase in SCFA-producing genera, such as *Clostridiales*, *Lachnospira*, *Roseburia*, *Lachnospiraceae*, and *Faecalibacterium*, following a 30 to 60 min exercise regimen over 6 weeks [87]. These findings support the relevance of exercise as a crucial component in the prevention of diseases associated with metabolic dysfunction.

### 5.3. Stress Management

As previously reported, various factors influence the GM composition, but especially the stress has shown a significant relationship with the gut–brain axis. This link between the microbiota and stress could affect the regulation of neural processes that are fundamental at every stage of development [88].

It has been observed that the gut microbiota and stress have a critical relationship impacting cognitive decline, showing an increase in the diversity of *Collinsella*, *Ruminococcus*, *Lactobacillus*, *Eubacterium*, *Mogibacterium*, *Desulfovibrio*, *Bulleidia*, *Succinivibrio*, *Corynebacterium*, and *Atopobium.* On the other hand, Faecalibacterium and *Turicibacter* showed a positive correlation with learning index scores and processing speed index [89].

Therefore, learning techniques for managing psychological stress and environmental stressors can be beneficial for the microbiota due to the close relationship between stress and dysbiosis, as a sustained state of stress has been shown to negatively impact the GM composition through the gut–brain axis [89,90].

### 5.4. Sleep Hygiene

Adequate quality and quantity of sleep are relevant for maintaining optimal health. Various studies have demonstrated that partial or prolonged sleep loss or disruption can alter the GM composition, promoting inflammation in adipose and systemic tissues. It has been observed that the relationship between *Firmicutes* and *Bacteroidetes* bacteria is affected, with a doubling of this ratio after two days of partial sleep deprivation compared to normal sleep [91].

Additionally, in young male patients, a positive correlation has been shown between the GM diversity and richness and sleep quality, as well as a negative correlation between microbiome diversity and sleep fragmentation [92]. The underlying mechanism of this microbiota–sleep relationship is not yet fully understood; various studies suggest that the gut microbiota may regulate this axis bidirectionally [93].

## 6. GM Modulation of the Gut Microbiota and Its Impact on Metabolic Health

### 6.1. Probiotics, Prebiotics, and Synbiotics

The administration of probiotics, prebiotics, and synbiotics could be a good approach for modulating the gut microbiota and improving metabolic health. Probiotics, defined by ISAPP (International Scientific Association for Probiotics and Prebiotics) as “live microorganisms which, when administered in adequate amounts, confer a health benefit to the host”, have shown positive effects on the intestinal mucosal barrier and the immune system. Clinical studies have revealed their benefits in patients with insulin resistance or dyslipidemias, suggesting their potential to prevent metabolic diseases such as type 2 diabetes mellitus and cardiovascular diseases associated with atherosclerosis [94].

Although probiotics are gaining recognition, their prophylactic use in metabolic diseases is still in the early stages of research. It is essential to consider the different species and amounts of probiotics to evaluate their potential preventive benefit [95].

Recent studies suggest that probiotics, prebiotics, and synbiotics are recognized for their effectiveness in treating MAFLD, including NAFLD and NASH. Guidelines emphasizing lifestyle changes—such as weight reduction, increased physical activity, and healthy diets—play a crucial role in managing NAFLD/NASH [96].

An imbalanced intestinal microbiome is associated with the pathogenesis of NAFLD/NASH. This GM dysbiosis can lead to alterations in the intestinal barrier and increased permeability, contributing to fat accumulation and, consequently, liver inflammation. This has directed therapy toward the intestinal microbiome as a novel therapeutic option. By modulating the intestinal environment, probiotics and synbiotics can reduce liver damage, improve metabolism, and decrease inflammation [94]. However, there is evidence that probiotics produce intestinal mucosal barriers, demonstrating their effect on the immune system [93], one of the reported mechanisms, as well as increasing mucin synthesis through goblet cells.

Meta-analyses that address the effects of traditional probiotics in patients with NAFLD/NASH have demonstrated favorable therapeutic outcomes, including attenuation of inflammatory mediators, modulation of lipid metabolism, improvement in liver fibrosis, facilitation of weight control, and obesity control [92].

Prebiotics, defined by ISAPP as “substrates that are selectively utilized by host microorganisms that confer a health benefit”, have been shown to influence the synthesis of glucagon-like peptide 1 and 2 (GLP-1 and GLP-2) in the proximal colon, as well as reduce fat mass, increase muscle mass, and improve glucose and lipid metabolism. This can provide protection against metabolic diseases.

Prebiotics, such as indigestible carbohydrates like glucans and fructans, positively impact colonic bacteria associated with health [96]. For example, they increase the number of *Lactobacillus* and *Bifidobacteria*, decrease the presence of the uremic toxin p-cresol and its serum concentration, improve lipid levels, oxidative stress indicators, and contrast systemic inflammation [90]. Both prebiotics and probiotics modulate the gut microbiota, promoting anaerobic bacterial metabolism and decreasing the production of solutes in hosts, including SCFA metabolism, bile salts, and metabolic endotoxemia [93,94].

Finally, synbiotics, defined by ISAPP as “a mixture comprising live microorganisms and substrates selectively utilized by host microorganisms that confer a health benefit”.

Synbiotics can further promote beneficial gut bacteria and the production of short-chain fatty acids (SCFAs), which have anti-inflammatory properties. This approach addresses some of the underlying factors that drive the progression of MAFLD and NASH [95,96].

Studies carried out in animal models showed improved glucose metabolism, better lipid biomarker profiles, reduced oxidative stress, prevention of a leaky gut phenotype, decreased serum lipopolysaccharides, and modulation of inflammatory, lipid, and glucose metabolism genes, along with restoration of the histomorphology of adipose tissue, the colon, and the liver [95].

Recent studies report that probiotics/synbiotics can improve transaminase levels, hepatic steatosis, and the NAFLD activity score. To some extent, probiotics/synbiotics can also reduce pro-inflammatory cytokines such as TNF-α and the interleukin family (IL-1, IL-6, IL-8) [97,98].

However, the future of probiotic research is challenged by a mix of personal beliefs, commercial interests, and insufficient medical regulation, which hinder objective interpretation. Despite this, advancements in microbiome research and new sequencing technologies offer the potential to move from empirical, one-size-fits-all approaches to targeted, patient-specific therapies. This shift requires a mechanism-focused approach that takes into account the population, medical conditions, and strain-specific effects. Key areas of focus include overcoming colonization resistance, understanding the long-term effects of probiotics, and developing personalized treatments. Large-scale, unbiased clinical trials free from commercial influence are crucial for establishing evidence-based guidelines and ensuring safety. Improved regulation is also necessary for the development of next-generation probiotics [99].

### 6.2. Fecal Microbiota Transplant (FMT)

It is known that fecal transplantation has been performed for a couple of decades now, involving the introduction of a suspension of fecal matter from a donor with specific characteristics into another person, with the aim of restoring bacterial groups that have been reduced or missing due to various causes, including specific diseases [100].

Today, FMT is a targeted therapeutic strategy used to treat and prevent non-communicable diseases, including metabolic and gastrointestinal conditions. This is based on certain bacterial communities that positively correlate with gastric motility, significantly impacting the improvement in metabolic dysregulation [101].

In T2DM patients, FMT has been performed, showing improvements in health. Results have revealed that these patients experienced an increase in GM diversity from the treatment start. This greater diversity has been positively associated with clinical values such as the homeostatic model assessment of insulin resistance (HOMA-IR) and BMI, as well as fasting glucose levels, postprandial glucose, and glycosylated hemoglobin [102].

Thus, it is suggested that the gut microbiota could be considered a therapeutic target for treating or preventing metabolic dysfunction through fecal transplantation [101].

Regarding the limitations, although FMT is a promising therapy for dysbiosis-related diseases, it has limitations and risks that must be carefully considered before use. There is a need for standardized protocols to ensure patient safety, taking into account relative contraindications, such as altered anatomy and anesthesia-associated risks with colonoscopy, physician factors (e.g., gastroenterologists vs. infectious disease specialists), and procedure-related risks, including the appropriateness of sedation or anesthesia. Additionally, thorough donor screening for viruses such as cytomegalovirus (CMV) and Epstein–Barr virus (EBV) is essential [102].

Severe risks, such as bacteremia or sepsis, have been associated with FMT due to insufficient screening for pathogens like Shiga toxin-producing *Escherichia coli* (STEC) or multidrug-resistant organisms (MDROs), including extended-spectrum beta-lactamase (ESBL)-producing bacteria, methicillin-resistant *Staphylococcus aureus* (MRSA), or carbapenem-resistant *Enterobacteriaceae* (CRE) [102].

## 7. Pharmacomicrobiomics in Metabolic Disorders

Pharmacomicrobiomics, an emerging field, highlights the crucial GM role in drug responses. Studies in this area have demonstrated that intestinal microorganisms and their metabolites can affect the bioavailability, clinical efficacy, and toxicity of a variety of drugs, both directly and indirectly [103].

This new field of pharmacomicrobiomics promises to simplify personalized medicine approaches based on the microbiome in various diseases by defining it as a proactive, well-coordinated, and well-tested structural strategy for effective medical care [104]. This is particularly facilitated by a network of electronic medical records that combine clinical and molecular data to offer the best possible treatment options, thereby enabling the development of patient-centered care for those who do not adequately respond to medications. Additionally, to improve therapeutic outcomes and mitigate medication side effects, various approaches have been suggested to selectively manipulate the microbiota, including the administration of probiotics and prebiotics alongside conventional treatments [105].

The interaction between pharmacomicrobiomics and the gut microbiota becomes evident in patients with T2DM who are treated with metformin, as significant alterations in their bacterial composition have been observed compared to those without treatment. Various animal models have explored the potential mechanisms behind this relationship, highlighting the GM influence on glucose metabolism, the regulation of SCFAs, and the modulation of the immune response [106,107]. These findings underscore the relevance of understanding how metformin affects metabolism through the gut microbiota, which could lead to the development of specific dietary interventions to optimize therapeutic outcomes and minimize side effects in T2DM patients [108].

The influence of pharmacomicrobiomics on health is also evident in hypertension treatment, where significant changes in GM composition have been observed in response to the angiotensin receptor antagonist, losartan. Studies conducted in spontaneously hypertensive rats (SHRs) treated with losartan have documented improvements in intestinal dysbiosis, characterized by reductions in *Firmicutes/Bacteroidetes* ratios and the presence of acetate and lactate-producing bacteria. Additionally, an increase in strict anaerobic bacteria has been observed, suggesting a shift towards a healthier microbiota. These GM changes have been associated with increased colonic integrity and decreased sympathetic activity in the gut, which may partly explain the antihypertensive effects of losartan [109]. These findings suggest that losartan-induced modulation of the gut microbiota contributes to vascular protection and blood pressure reduction, possibly through the regulation of the immune system in the gut.

### Adequate Use of Medications with a Described Role in GM Modulating the Gut Microbiota

It is crucial to recognize that multiple medications can have negative effects on the gut microbiota, which can have long-term health consequences. For example, antibiotics kill pathogenic bacteria but also beneficial ones in the gut, decreasing bacterial diversity and the relative abundance of beneficial genera such as *Bifidobacterium* and *Lactobacillus* [110].

Similarly, medications such as nonsteroidal anti-inflammatory drugs (NSAIDs) and proton pump inhibitors (PPIs) show a negative impact on GM, promoting intestinal dysbiosis as a consequence of their use and/or negatively affecting the production of SCFAs [111].

In this regard, it is relevant to emphasize that the goal is not to be against the use of these medications but to promote their appropriate prescription and avoid overmedication. Healthcare professionals should ensure that only necessary medications are prescribed at the correct dosage, avoiding unnecessary or excessive prescription of antibiotics and other medications that could affect GM composition and/or function [109,110].

Additionally, it is critical to raise awareness among patients about the relevance of avoiding self-medication and informing their medical doctor about any medications they are taking. Patients should also be aware of the potential side effects of medications and how these may impact their GM and overall health [112].

## 8. Discussion

The bacterial profile of an individual could be closely linked to their health, especially through the presence of a balanced microbiota. However, the mechanisms by which disruptions in the microbiome, referred to as dysbiosis, contribute to the development of metabolic diseases remain unclear. In addition, recent studies suggest that host susceptibility to these diseases is influenced by their immune-modulatory responses. This suggests that factors beyond genetics play a significant role. Finally, lifestyle choices, diet, and geographic location can all modify bacterial profiles.

As a result, studying the microbiome offers valuable insights into the understanding of non-communicable diseases, highlighting the relevance of a holistic approach to health.

In recent decades, research has focused on trying to understand the origin of non-communicable diseases from a dynamic point of view; therefore, an attempt has been made to relate the GM influence on the therapeutic effects of drugs since the microbiota and its metabolites could modify the pharmacokinetics and pharmacodynamics of these drugs, decreasing or increasing their effectiveness [113].

Orally administered drugs pass through the upper gastrointestinal tract and the small intestine before reaching the large intestine, where they come into contact with thousands of microorganisms, which can synthesize various enzymes involved in drug metabolism, including oxidases and hydrogenases [114,115].

Research shows that drugs can influence the host’s metabolism by modulating the GM composition or function, with many of these changes leading to dysbiosis. For instance, the intestinal microbiota can enhance the bioavailability of some medicines, such as nifedipine, by affecting its metabolism and therapeutic efficacy. Additionally, captopril treatment may help balance the dysfunctional gut–brain axis in male offspring by altering the gut microbiota, thereby improving intestinal health and permeability.

Metformin has been shown to increase the abundance of *Akkermansia muciniphila* in the intestine, promoting the production of SCFAs and positively affecting insulin resistance and glucose homeostasis [116]. Conversely, proton pump inhibitors can alter the composition of the intestinal microbiota by modifying gastrointestinal pH and slowing gastric emptying.

Sequencing-based microbiome studies have resulted in significant advances and discoveries regarding the microbiome and the impact of specific bacterial species or genera. However, it is critical to acknowledge the limitations associated with microbiota analysis. Variations in different sequencing techniques, along with the selection of specific regions for analysis, can pose challenges, especially in achieving high sensitivity for the taxa detection and quantification [117]. Additionally, the lack of standardized techniques complicates efforts to establish a gold standard for microbiome analysis

## 9. Conclusions

Currently, it is known that the onset of chronic non-communicable diseases (such as metabolic syndrome, diabetes, dyslipidemia, and inflammatory diseases, among others) is promoted by multiple factors that are crucial for an adequate response to various treatments. Therefore, multiple health strategies have been sought to prevent or control these disorders. Research tools have allowed us to elucidate certain mechanisms that could support improvements in the quality of life for the population.

Today, the gut microbiota has become a major focus of interest because it has been observed that depending on the bacterial diversity present in an individual’s gastrointestinal tract, there may be greater susceptibility to modifying or reversing certain patterns that over time become harmful to the organism, potentially leading to intestinal dysbiosis and associated symptoms.

A balanced diet rich in elements such as fiber, polyphenolic compounds, and omega-3 fatty acids is crucial for optimal GM health. Therefore, it is needed to adopt an appropriate dietary approach, which should be personalized according to individual needs, emphasizing the addition of nutrients that provide significant nutritional value.

In addition, it is essential to consider other factors to achieve a positive long-term response. Thus, various strategies can be implemented for improvement, including good nutrition, physical exercise, stress management, and proper sleep hygiene. These elements not only have a positive impact on GM but can also provide additional benefits, enhancing the quality of life for those who implement these interventions.

Furthermore, several aspects should be considered from the perspective of prevention and treatment, such as the use of probiotics, prebiotics, symbiotics, and fecal transplants to modulate the gut microbiota. These have been shown to positively impact intestinal health, decreasing systemic inflammation, and improving lipid levels. They also demonstrate a synergistic effect by restoring adipose tissue, the colon, and the liver, and modulating inflammatory, lipid, and glucose metabolism.

It is relevant to highlight the impact that the diagnosis could have from knowing the bacterial profile coming from the intestine. This will allow us to achieve a timely and accurate diagnosis in addition to being assertive with the treatment, which is tailored to each individual’s specific needs.

There is a need for future research to better understand the GM role, as there are still relevant limitations to address. While the currently used adjuvant therapies, which involve specific bacterial colonies from healthy donors or supplements, have been shown to be safe in the short term, various risks may arise. These include issues related to donor availability and ensuring that the selected bacterial profile is appropriate for treating the specific non-communicable disease.

In cases where treatment is based on a supplement, it is essential that the product meets established standards and demonstrates proven effectiveness. This includes ensuring an appropriate abundance and diversity of species that can effectively repopulate the gut niche, which has shown positive health effects. Additionally, it is needed to consider that the costs of these therapies can sometimes be high.

In this field, there are multiple aspects that remain to be investigated due to deficiencies in both research and clinical practices that support personalized treatment. It is correct to recognize some limits in the comprehensive study of the intestinal microbiota, which opens up diverse possibilities for further exploration. This ongoing research can lead to new methodologies, diagnostic strategies, and treatment options for studying and understanding the microbiome. Additionally, it enhances the knowledge of healthcare personnel on the front lines of patient care.

Finally, it is relevant to stress that the gut microbiota is a complex system, and there is still much to learn about its functioning and its relationship with the development of diseases.

## Figures and Tables

**Figure 1 biomedicines-12-02529-f001:**
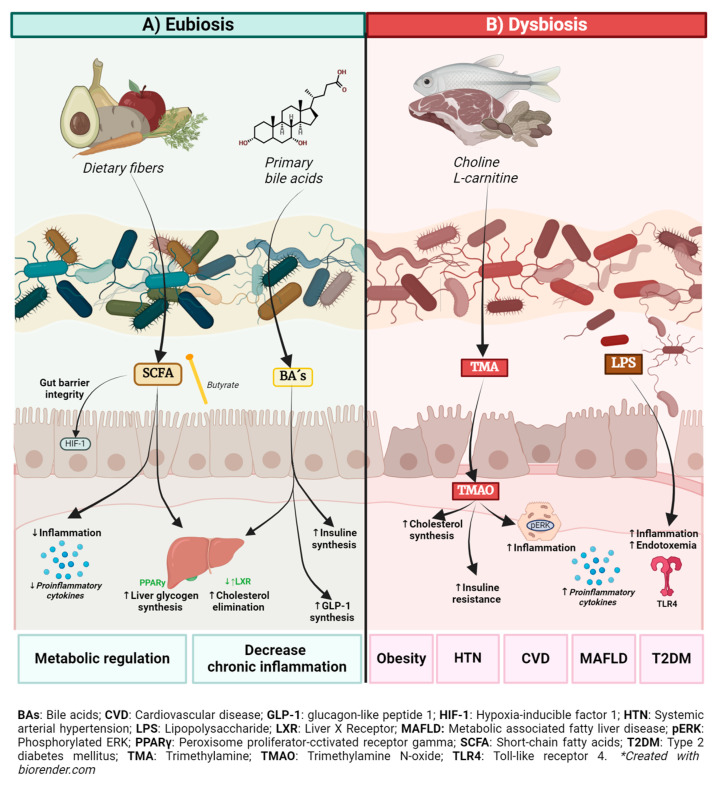
Gut microbiota (GM) in eubiosis and dysbiosis and associated metabolites. (**A**) In a state of eubiosis; blue box, GM is characterized by a diverse and rich bacterial community that the increase of beneficial metabolites, like short-chain fatty acids (SCFAs). These metabolites have anti-inflammatory properties, regulate cholesterol and glucose metabolism and provide energy and maintain the integrity of the gut barrier, reducing the risk of degenerative diseases and preserving the gut barrier. (**B**) The figure also illustrates the consequences of intestinal disbiosis; red box, the loss of gut permeability, caused by an imbalance in microbial composition and bacterial translocation. This state is marked by an increase in levels of harmful metabolites, such as trimethylamine N-oxide (TMAO) and lipopolysaccharides (LPS), which contribute to increased intestinal permeability, endotoxemia, and chronic inflammation. Dysbiosis also activates toll-like receptor 4 (TLR4), triggering an inflammatory cytokine cascade. Elevated TMAO and LPS levels have been linked to insulin resistance, inflammation, and a higher risk of metabolic disorders, including obesity, diabetes type 2 (T2DM), systemic arterial hypertension, and cardiovascular diseases (CVD).

## Data Availability

Not applicable.

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
