# Peer review of "Primary Prevention Strategy for Non-Communicable Diseases (NCDs) and Their Risk Factors: The Role of Intestinal Microbiota"

_biomedicines, 2024, doi:10.3390/biomedicines12112529_

Round 1
Reviewer 1 Report
Comments and Suggestions for Authors
1. The flow could be improved by reducing repetition and ensuring smoother transitions between topics.
2. Include a section on research gaps and the potential for genome-wide microbiome studies to predict NCD risk.
3. The review should differentiate itself more clearly from other similar reviews by focusing on underexplored areas, such as the role of gut microbiota in less commonly discussed NCDs (e.g., neurological diseases, autoimmune disorders).
4. Integrating insights from immunology, endocrinology, and systems biology could provide a more holistic understanding of how gut microbiota interact with various body systems, especially in NCD progression.
5. More attention to the variation in microbiota composition across different populations, particularly in relation to genetic, dietary, and environmental factors, would enhance the relevance of the review for a global audience
6. Including a discussion on how gut microbiota can influence the metabolism of medications commonly used for NCDs (e.g., statins, metformin) could provide a unique angle not commonly covered in similar reviews
7. The review should address the limitations of current microbiome research methods, including biases in sequencing technologies and the lack of standardized protocols, which could influence the interpretation of findings.
8. Address ethical, regulatory, and accessibility challenges related to microbiota-based interventions.
9. Strengthen the conclusion by emphasizing the need for integrating microbiota findings into public health strategies.
10. These additional points would help deepen the review and make it stand out more in the crowded space of microbiome and NCD-related literature.
Author Response
Q1: The flow could be improved by reducing repetition and ensuring smoother transitions between topics.
A1: I would like to thank the reviewer for his point of view as it is very relevant. The manuscript has been exhaustively reviewed reducing repetition and ensuring smoother transitions between topics we considered the reviewer suggestions that improve flow and ensure a better understanding.
Q2: Include a section on research gaps and the potential for genome-wide microbiome studies to predict NCD risk.
A2: We thank the reviewer for the critical and adequate suggestion. It has been included in the discussion section on lines 640-648.
The bacterial profile of an individual could be closely linked to their health, especially through the presence of a balanced microbiota. However, the mechanisms by which disruptions in the microbiome, referred to as dysbiosis, contribute to the development of metabolic diseases remain unclear. In addition, recent studies suggest that host susceptibility to these diseases is influenced by their immune-modulatory responses. This suggests that factors beyond genetics play a significant role. Finally the lifestyle choices, diet, and geographic location can all modify bacterial profiles.
As a result, studying the microbiome offers valuable insights into the understanding of non-communicable diseases, highlighting the relevance of a holistic approach to health.
118.- Sirisinha S. The potential impact of gut microbiota on your health:Current status and future challenges. Asian Pac J Allergy Immunol. 2016 Dec;34(4):249-264.
Q3: The review should differentiate itself more clearly from other similar reviews by focusing on underexplored areas, such as the role of gut microbiota in less commonly discussed NCDs (e.g., neurological diseases, autoimmune disorders).
A3: (not included in manuscript)
I appreciate the reviewer interest in highlighting the manuscript's focus on lesser-studied diseases, which are surely relevant. However, the field of non-communicable diseases is particularly significant due to their high prevalence worldwide and their associated morbidity and mortality. This highlights the potential impact that the microbiota can have on health and disease within the population.
2.- Global Burden of Disease Collaborative Network, Global Burden of Disease Study 2019 (GBD 2019) Results (2020, Institute for Health Metrics and Evaluation – IHME)
Q4: Integrating insights from immunology, endocrinology, and systems biology could provide a more holistic understanding of how gut microbiota interact with various body systems, especially in NCD progression.
A4: We thank the reviewer for the critical and appropriate suggestion (not included)
The intestinal microbiota plays a crucial role in protecting the organism, and understanding the disease progression requires acknowledging the complexity of these interactions, which are not solely unidirectional. At the systemic level, there are different control points designed to maintain cellular homeostasis. When this balance is disrupted, it leads to changes in the bacterial composition. This dysbiosis can compromise the epithelial barrier, increasing its permeability and exposing the body to molecular patterns associated with pathogens and lipopolysaccharides1. Consequently, harmful bacteria can proliferate, leading to inflammation2.
Furthermore, some metabolites produced during dysbiosis are linked to hormonal changes3, favoring an endocrine imbalance. This disruption not only alters the immune response but also heightens inflammation, triggering the production of additional immune molecules. As a result, the individuals may become more susceptible to various metabolic disorders, which can contribute to a chronic inflammatory process affecting all biological systems3.
1.- Ferreira RDS, Mendonça LABM, Ribeiro CFA, Calças NC, Guimarães RCA, Nascimento VAD, Gielow KCF, Carvalho CME, Castro AP, Franco OL. Relationship between intestinal microbiota, diet and biological systems: an integrated view. Crit Rev Food Sci Nutr. 2022;62(5):1166-1186.
2.- Yoo JY, Groer M, Dutra SVO, Sarkar A, McSkimming DI. Gut Microbiota and Immune System Interactions. Microorganisms. 2020 Oct 15;8(10):1587. doi: 10.3390/microorganisms8101587. Erratum in: Microorganisms. 2020 Dec 21;8(12):E2046.
3.- Liu Q, Sun W, Zhang H. Interaction of Gut Microbiota with Endocrine Homeostasis and Thyroid Cancer. Cancers (Basel). 2022 May 27;14(11):2656.
Q5: More attention to the variation in microbiota composition across different populations, particularly in relation to genetic, dietary, and environmental factors, would enhance the relevance of the review for a global audience.
A5: We thank the reviewer for the right suggestion (not included)
I appreciate the reviewer’s suggestion, especially regarding the significant variations in intestinal microbiota among individuals from the same family. Each person has an unique microbial fingerprint that distinguishes them from others. Moreover, the intestinal microbiota can be modified in terms of species diversity and abundance through specific daily changes in habits, including environmental factors, ethnicity, gender, age, diet, lifestyle, exercise, and body composition (BMI) (doi: 10.1016/j.celrep.2021.109765). Numerous studies conducted in various populations, primarily in Europe and Asia, documented these differences in microbiome profiles.
Q6: Including a discussion on how gut microbiota can influence the metabolism of medications commonly used for NCDs (e.g., statins, metformin) could provide a unique angle not commonly covered in similar reviews
A6: I appreciate your relevant contribution. This part is therefore included in the discussion section lines (649-667)
In recent decades, research has focused on trying to understand the origin of non-communicable diseases, from a dynamic point of view, therefore, an attempt has been made to relate the GM influence on the therapeutic effects of drugs, since the microbiota its metabolites could modify the pharmacokinetics and pharmacodynamics of these drugs, decreasing or increasing their effectiveness118.
Orally administered drugs pass through the upper gastrointestinal tract and the small intestine before reaching the large intestine, where they come into contact with thousands of microorganisms whichcan synthesize various enzymes involved in drug metabolism, including oxidases and hydrogenases119.
Research shows that drugs can influence the host's metabolism by modulating the GM composition or function, with many of these changes leading to dysbiosis. For instance, the intestinal microbiota can enhance the bioavailability of some medicines, such as nifedipine, by affecting its metabolism and therapeutic efficacy. Additionally, captopril treatment may help balance the dysfunctional gut-brain axis in male offspring by altering the gut microbiota, thereby improving intestinal health and permeability.
Metformin has been shown to increase the abundance of Akkermansia muciniphila in the intestine, promoting the production of SCFAs and positively affecting insulin resistance and glucose homeostasis 120. Conversely, proton pump inhibitors can alter the composition of the intestinal microbiota by modifying gastrointestinal pH and slowing gastric emptying.
118.- Wu H, Esteve E, Tremaroli V, Khan MT, Caesar R, Mannerås-Holm L, et al. Metformin alters the gut microbiome of individuals with treatment-naive type 2 diabetes, contributing to the therapeutic effects of the drug. Nat Med. (2017) 23:850–8.
119.- Doestzada M, Vila AV, Zhernakova A, Koonen DPY, Weersma RK, Touw DJ, et al. Pharmacomicrobiomics: a novel route towards personalized medicine? Protein Cell. (2018) 9:432–45.
120.- Forslund K, Hildebrand F, Nielsen T, Falony G, Le Chatelier E, Sunagawa S, et al. Disentangling type 2 diabetes and metformin treatment signatures in the human gut microbiota. Nature. (2015) 528:2626.
Q7: The review should address the limitations of current microbiome research methods, including biases in sequencing technologies and the lack of standardized protocols, which could influence the interpretation of findings.
A7: I appreciate your important contribution. This part is therefore included in the discussion section lines (668-674)
Sequencing-based microbiome studies have resulted in significant advances and discoveries regarding the microbiome and the impact of specific bacterial species or genera. However, it is critical to acknowledge the limitations associated with microbiota analysis. Variations in different sequencing techniques, along with the selection of specific regions for analysis, can pose challenges, especially in achieving high sensitivity for the taxa detection and quantification. Additionally, the lack of standardized techniques complicates efforts to establish a gold standard for microbiome analysis
122.- Pinto Y, Bhatt AS. Sequencing-based analysis of microbiomes. Nat Rev Genet. 2024 Jun 25.
Q8: Address ethical, regulatory, and accessibility challenges related to microbiota-based interventions.
A8: We thank the reviewer for the appropriate suggestion (not included)
Regarding the adjuvant therapies currently in use, especially those involving specific bacterial colonies from healthy donors or supplements, these procedures have been shown to be safe in the short term. However, there are potential risks associated with the treatment, including issues related to donor availability and ensuring that the bacterial profile selected is suitable for addressing the specific non-communicable disease.
A significant concern with fecal microbiota transplantation (FMT) is the potential transfer of antibiotic resistance genes from the donor's feces to the recipient, which can complicate treatment outcomes. On the other hand, the use of supplements that meet established standards has demonstrated a positive effect on health, especially in terms of the abundance and diversity of species that repopulate the gut niche. Nonetheless, it is right to consider that the costs of these therapies can be quite high.
Q9: Strengthen the conclusion by emphasizing the need for integrating microbiota findings into public health strategies.
A9: Thank you for your important suggestion, which was already included in the conclusions section lines 701-704
It is relevant to highlight the impact that the diagnosis could have from knowing the bacterial profile coming from the intestine. This will allow to achieve a timely and accurate diagnosis in addition to being assertive with the treatment, which is tailored to each individual’s specific needs.
Q10: These additional points would help deepen the review and make it stand out more in the crowded space of microbiome and NCD-related literature.
A10: Your comments were extremely relevant, since you have provided suggestions that show an improvement in the quality of the manuscript, for which I thank you for the time dedicated to reviewing it.

Reviewer 2 Report
Comments and Suggestions for Authors
I have read Lopez-Tenorio et al.'s study entitled "Primary prevention strategy for non-communicable diseases (NCDs) and their risk factors: the role of intestinal microbiota". Please find below my comments for revision. These comments aim to improve the study's scientific rigor, transparency, and comprehensive understanding of the findings:
1. Provide a detailed explanation of microbial shifts leading to disease, including how specific bacteria influence metabolic pathways and clinical outcomes.
2. Clarify the role of TMAO in metabolic diseases and include more evidence on its impact on disease progression.
3. Acknowledge the potential for different SCFAs to have varying effects on inflammation in diverse metabolic states.
4. Strengthen the link between animal studies and human MAFLD by discussing the translatability of fecal transplantation findings and how gut microbiota contribute to liver inflammation and steatosis in humans.
5. Provide more detail on how changes in microbial composition impact metabolic pathways and the progression of MAFLD.
6. Elaborate on how gut-derived products like LPS lead to systemic inflammation and insulin resistance using updated human studies.
7. Expand on specific mechanisms of how microbial metabolites influence sleep via the gut-brain axis and include hormonal pathways to better understand the bidirectional relationship.
8. Provide a critical evaluation of clinical evidence on probiotics, highlighting the mixed results from human trials and emphasizing the need for personalized approaches.
9. Address the risks and limitations of FMT, such as pathogen transmission and donor variability, while noting the lack of long-term data on its efficacy in metabolic disorders.
10. Discuss and compare your findings with recent publications with similar topics, such as PMC9301375.
Author Response
Reviewer 2
Q1: Provide a detailed explanation of microbial shifts leading to disease, including how specific bacteria influence metabolic pathways and clinical outcomes.
A1: We thank the reviewer for the critical suggestion (not included)
Healthy gut microbiome is predominantly characterized by bacteria from the phyla Bacteroidetes, Firmicutes, and Actinobacteria, along with high microbial diversity. When there is an overgrowth of Proteobacteria, the expansion of new bacterial groups, or decreased microbial diversity, dysbiosis occurs. As previously reported, it can arise from multiple causes, such as a diet rich in fats, complex carbohydrates, low fiber, or the misuse of antibiotics or non-steroidal anti-inflammatory drugs, as seen in Clostridium difficile-associated diarrhea. (1)
Multiple diseases are linked to intestinal dysbiosis, which has an inflammatory and immune-related basis. First, the translocation of bacterial products like lipopolysaccharides (LPS) activates toll-like receptors (TLRs), leading to the overexpression of pro-inflammatory cytokines, which damage the gut epithelium and contribute to chronic inflammation. Additionally, a damaged microbiota affects the immunity maturation, especially innate components . Without a healthy microbiota, the functions of neutrophils and dendritic cells (DCs) are impaired, resulting in reduced pathogen-killing ability and decreased secretion of type I interferons (IFN-I) and interleukin-15 (IL-15), respectively. Persistent alterations caused by antibiotic treatment in early life have been linked to the development of inflammatory bowel disease (IBD), asthma, and atopic dermatitis later in life. (1)
- Weiss GA, Hennet T. Mechanisms and consequences of intestinal dysbiosis. Cell Mol Life Sci. 2017 Aug;74(16):2959-2977.
Q2: Clarify the role of TMAO in metabolic diseases and include more evidence on its impact on disease progression.
A2: We thank the reviewer for the critical suggestion. In agreement, we have better described it in Section 2.3 (please see the lines 140-161). We hope to have addressed all expectations from the reviewer.
In recent years, trimethylamine N-oxide (TMAO) has emerged as a metabolite of great interest due to its association with inflammatory diseases, especially cardiovascular diseases.
TMAO is formed in the liver from trimethylamine (TMA), which is produced by the gut microbiota (GM) from phosphatidylcholine and L-carnitine found in dietary foods such as red meat, cheese, and eggs 25. The enzyme flavin-containing monooxygenase 3 (FMO3) catalyzes the conversion of TMA to TMAO. This TMA/FMO3/TMAO axis has been associated with increased insulin resistance and hepatic lipogenesis 25-26. The TMAO levels in blood are elevated in consequence of gut dysbiosis, since the members of the Firmicutes, Actinobacteria and Proteobacteria phyla, that main producer of TMA precursor, are characteristic of gut dysbiosis.
TMAO can trigger inflammation pathways, increasing the expression IL-1β and IL-6 through the activation of NF-κB. This results in a systemic proinflammatory state that can trigger or worsen diseases where inflammation plays a significant role, such as atherosclerosis, diabetes, and hypertension.
For example, elevated blood TMAO levels have been observed in some patients with acute myocardial infarction 27. Specifically, ST-elevated myocardial infarction (STEMI) is characterized by elevated levels of IL-6, IL-1β and C-reactive protein. Elevated levels of TMAO are considered as risk factor for STEMI.
In diabetes, it has been found that higher TMAO levels are in diabetes patients, in comparison with prediabetes and non-diabetes patients. It has been reported that deletion if FMO3, an enzyme responsible for TMA to TMAO conversion, increases glucose levels and insulin resistance (27-28)..
Q3: Acknowledge the potential for different SCFAs to have varying effects on inflammation in diverse metabolic states.
A3: We thank the reviewer for the constructive suggestion. In agreement, we have better described Section 2.2 (please see the lines 124-138).
GM metabolizes complex dietary carbohydrates through fermentation, producing SCFAs, such as acetate, propionate, and butyrate.20 These SCFAs are primarily produced by bacterial families like Firmicutes, Bacteroidetes, and other anaerobes.21 SCFAs play a crucial role in modulating host metabolic pathways, exhibiting anti-inflammatory effects, and serving as an energy source for intestinal epithelial cells.22,23 For example, butyrate activates hypoxia-inducible factor 1 (HIF-1), promoting gene expression like erythropoietin (EPO), improving intestinal barrier integrity, and inhibiting nuclear factor kappa B (NF-kB) activity, thus reducing pro-inflammatory cytokine production.24
Recent studies suggest that SCFAs production varies with the host’s metabolic state. In patients with obesity or type 2 diabetes, reduced SCFAs has been linked to increased low-grade inflammation associated with these metabolic states. However, other studies have found elevated SCFA levels in the intestines of obese patients, suggesting a paradoxical effect where SCFAs may contribute both to metabolic regulation and to energy accumulation.25 Hence, SCFAs play a dual role: while their production benefits inflammatory balance under normal conditions, excessive accumulation may be involved in obesity and insulin resistance.
- Lee CJ, Sears CL, Maruthur N. Gut microbiome and its role in obesity and insulin resistance. Ann N Y Acad Sci. 2020 Feb;1461(1):37-52. doi: 10.1111/nyas.14107. Epub 2019 May 14. PMID: 31087391.
Q4: Strengthen the link between animal studies and human MAFLD by discussing the translatability of fecal transplantation findings and how gut microbiota contribute to liver inflammation and steatosis in humans.
A4: We thank the reviewer for appropriate suggestion (not included)
In humans, FMT studies have shown that transferring gut microbiota from lean donors to obese recipients improves insulin sensitivity and reduces liver inflammation. Similarly, synbiotic treatments in MAFLD patients have demonstrated beneficial effects, including reduced liver steatosis, lower inflammation markers like ALT and TNF-α, and an enrichment of beneficial gut bacteria (Lactobacillus, Bifidobacterium, Faecalibacterium). These treatments also reduced gut transit time and enriched genera associated with improved metabolic health1. Anyways, further research is required to clarify the microbial mechanisms behind these effects.
Mitrović, M., Dobrosavljević, A., Odanović, O., Knežević-Ivanovski, T., Kralj, Đ., Erceg, S., Perućica, A., Svorcan, P., & Stanković-Popović, V. (2024). The effects of synbiotics on the liver steatosis, inflammation, and gut microbiome of metabolic dysfunction-associated liver disease patients-randomized trial. Romanian journal of internal medicine = Revue roumaine de medecine interne, 62(2), 184–193.
Q5: Provide more detail on how changes in microbial composition impact metabolic pathways and the progression of MAFLD.
A5: We thank the reviewer for the appropriate suggestion (not included).
Changes in the composition of the bacterial community can lead to intestinal dysbiosis, which has significant implications for the organism's overall balance. In detail, the epithelial barrier may be adversely affected, resulting in increased intestinal permeability due to exposure to molecular patterns associated with pathogens and the presence of lipopolysaccharides (LPS) in circulation1, along with the migration of circulating macrophages.
This dysbiosis subsequently activates lipid metabolism pathways, promoting alterations in lipolysis and adipogenesis, which result in the accumulation of adipose tissue in various organs, primarily the liver2. At the cellular level, there is an increase in reactive oxygen species, as well as elevated levels of pro-inflammatory cytokines such as IL-1β, IL-6, and TNF. Consequently, this pathological state triggers a chronic inflammatory process3.
1.- Ferreira RDS, Mendonça LABM, Ribeiro CFA, Calças NC, Guimarães RCA, Nascimento VAD, Gielow KCF, Carvalho CME, Castro AP, Franco OL. Relationship between intestinal microbiota, diet and biological systems: an integrated view. Crit Rev Food Sci Nutr. 2022;62(5):1166-1186.
2.- Yoo JY, Groer M, Dutra SVO, Sarkar A, McSkimming DI. Gut Microbiota and Immune System Interactions. Microorganisms. 2020 Oct 15;8(10):1587.
3.-Mitrović, M., Dobrosavljević, A., Odanović, O., Knežević-Ivanovski, T., Kralj, Đ., Erceg, S., Perućica, A., Svorcan, P., & Stanković-Popović, V. (2024). The effects of synbiotics on the liver steatosis, inflammation, and gut microbiome of metabolic dysfunction-associated liver disease patients-randomized trial. Romanian journal of internal medicine = Revue roumaine de medecine interne, 62(2), 184–193.
Q6: Elaborate on how gut-derived products like LPS lead to systemic inflammation and insulin resistance using updated human studies.
A6: We thank the reviewer for the right suggestion (not included in manuscript)
One of the most relevant metabolites involved in dysbiosis-related systemic inflammation is LPS,that can disrupt the intestinal barrier and enter the bloodstream, leading to metabolic endotoxemia and causing both systemic and local inflammation. It triggers an inflammatory response by binding to Toll-like receptor 4 (TLR4). When LPS circulates in the body, it interacts with LPS-binding protein (LBP), which is recognized by the membrane-bound co-receptor CD14. This interaction allows LPS to connect with the membrane protein MD-2 and the extracellular part of TLR4, activating signaling pathways that result in the production of pro-inflammatory cytokines. (1,2)
In vitro models show a range of reactions in response to intravenous LPS, from clinical outcomes such as elevated body temperature and tachycardia to biochemical changes like increased levels of TNF-α, interferon (IFN)-γ, IL-6, and IL-8. Additionally, epigenetic changes, such as histone modifications, lead to an open chromatin structure and increased transcription of genes, further amplifying the inflammatory response. (1)
This LPS-induced inflammation contributes to various metabolic diseases, including obesity and type 2 diabetes mellitus (T2DM). Studies have shown that plasma LPS levels are 76% higher in patients with T2DM compared to controls. In obesity and T2DM, LPS activates an immune response in adipose tissue. In patients with T2DM, NF-kB activation leads to the production of adipokines in adipose tissue. Treatment of human abdominal subcutaneous adipocytes with LPS significantly increases the secretion of pro-inflammatory cytokines, TNF-α and IL-6, which promotes insulin resistance and pancreatic β-cell dysfunction. (2)
1.- Brooks, D., Barr, L. C., Wiscombe, S., McAuley, D. F., Simpson, A. J., & Rostron, A. J. (2020). Human lipopolysaccharide models provide mechanistic and therapeutic insights into systemic and pulmonary inflammation. The European respiratory journal, 56(1), 1901298.
2.- Mohammad, S., & Thiemermann, C. (2021). Role of Metabolic Endotoxemia in Systemic Inflammation and Potential Interventions. Frontiers in immunology, 11, 594150.
Q7: Expand on specific mechanisms of how microbial metabolites influence sleep via the gut-brain axis and include hormonal pathways to better understand the bidirectional relationship.
A7: Excellent and interesting topic to address, we attach the comment related to it.
The relationship between the gut microbiome and the nervous system is well-known to be bidirectional, involving multiple pathways such as neurotransmitters, hormones, and metabolites that facilitate communication. (1) The gut has its own complex neuronal system, with the vagus nerve serving as the primary communication pathway to the central nervous system. Studies have established a link between gut dysbiosis and neuropsychiatric as well as neurodevelopmental disorders. (1,2) This connection has led to growing interest in the study of the gut microbiome's role in sleep disorders.
Sleep disturbances also have a bidirectional relationship with the gut microbiome. Sleep deprivation and fragmentation can compromise the intestinal barrier integrity , potentially leading to the leakage of bacterial metabolites. In addition, sleep quality and duration are associated with different bacterial profiles. For example, sleep dysfunction has been linked to changes in the abundance of taxa such as Paracoccus, Rikenellaceae, and Clostridium in fecal samples. (3,4)
Conversely, the gut microbiome can influence sleep through the production of metabolites, hormones, and neurotransmitters. Several key relationships have been identified: certain SCFAs like propionate, found in higher levels, have been associated with longer uninterrupted sleeping infants. Melatonin, which can be produced by gut microbiota, affects sleep by acting on MT1 and MT2 receptor signaling in the brain. The gut microbiome can also influence the metabolism of neurotransmitters like GABA, known to promote sleep, and serotonin, produced by bacterial families such as Clostridiaceae and Turicibacteraceae, which helps regulate sleep. (4)
1.- Wang, Q., Yang, Q., & Liu, X. (2023). The microbiota-gut-brain axis and neurodevelopmental disorders. Protein & cell, 14(10), 762–775.
2.- Góralczyk-Bińkowska, A., Szmajda-Krygier, D., & Kozłowska, E. (2022). The Microbiota-Gut-Brain Axis in Psychiatric Disorders. International journal of molecular sciences, 23(19), 11245.
3.- Matenchuk, B. A., Mandhane, P. J., & Kozyrskyj, A. L. (2020). Sleep, circadian rhythm, and gut microbiota. Sleep medicine reviews, 53, 101340.
4.- Wang, Z., Wang, Z., Lu, T., Chen, W., Yan, W., Yuan, K., Shi, L., Liu, X., Zhou, X., Shi, J., Vitiello, M. V., Han, Y., & Lu, L. (2022). The microbiota-gut-brain axis in sleep disorders. Sleep medicine reviews, 65, 101691
Q8: Provide a critical evaluation of clinical evidence on probiotics, highlighting the mixed results from human trials and emphasizing the need for personalized approaches.
A8: In agreement with the right suggestion, we have better described section (please see the lines 547-557)
Probiotics have been extensively researched over the years by scientists and the food and pharmaceutical industries, leading to various proposed health benefits. These include the prevention and treatment of conditions like acute diarrhea, antibiotic-associated diarrhea, Clostridium difficile–associated diarrhea, and the management of inflammatory bowel disease and irritable bowel syndrome (IBS). However, the future of probiotic research is challenged by a mix of personal beliefs, commercial interests, and insufficient medical regulation, which hinder objective interpretation. Despite this, advancements in microbiome research and new sequencing technologies offer the potential to move from empirical, one-size-fits-all approaches to targeted, patient-specific therapies. This shift requires a mechanism-focused approach that takes into account the population, medical conditions, and strain-specific effects. Key areas of focus include overcoming colonization resistance, understanding the long-term effects of probiotics, and developing personalized treatments. Large-scale, unbiased clinical trials free from commercial influence are crucial for establishing evidence-based guidelines and ensuring safety. Improved regulation is also necessary for the development of next-generation probiotics105.
105.- Suez J, Zmora N, Segal E, Elinav E. The pros, cons, and many unknowns of probiotics. Nat Med. 2019 May;25(5):716-729. doi: 10.1038/s41591-019-0439-x. Epub 2019 May 6. PMID: 31061539.
Q9: Address the risks and limitations of FMT, such as pathogen transmission and donor variability, while noting the lack of long-term data on its efficacy in metabolic disorders
A9: (please see the lines 574-586)
Regarding the limitations, although FMT is a promising therapy for dysbiosis-related diseases, it has limitations and risks that must be carefully considered before use. There is a need for standardized protocols to ensure patient safety, taking into account relative contraindications, such as altered anatomy and anesthesia-associated risks with colonoscopy, physician factors (e.g., gastroenterologists vs. infectious disease specialists), and procedure-related risks, including the appropriateness of sedation or anesthesia. Additionally, thorough donor screening for viruses such as cytomegalovirus (CMV) and Epstein-Barr virus (EBV) is essential108.
Severe risks, such as bacteremia or sepsis, have been associated with FMT due to insufficient screening for pathogens like Shiga toxin-producing Escherichia coli (STEC) or multidrug-resistant organisms (MDROs), including extended-spectrum beta-lactamase (ESBL)-producing bacteria, methicillin-resistant Staphylococcus aureus (MRSA), or carbapenem-resistant Enterobacteriaceae (CRE) 108.
- - Gupta S, Mullish BH, Allegretti JR. Fecal Microbiota Transplantation: The Evolving Risk Landscape. Am J Gastroenterol. 2021 Apr;116(4):647-656.
Q10: Discuss and compare your findings with recent publications with similar topics, such as PMC9301375.
A10: We thank the reviewer for the critical and appropriate suggestion. We hope to have satisfy the reviewer expectations.
Several authors have emphasized the critical role of the gut microbiome in the development of non-communicable diseases (NCDs), highlighting the mechanisms by which dysbiosis—an imbalance in gut microbial populations—can contribute to diseases such as type 2 diabetes (T2D), cardiovascular disease, and obesity. They propose that these mechanisms could explain how a previously healthy individual might develop NCDs after receiving a fecal microbiota transplant (FMT) from a donor with a diseased microbiome, suggesting the possibility that NCDs could, in certain cases, become "communicable" through microbiome transmission. (1)
This hypothesis underscores the importance of focusing on the gut microbiome in NCD research to develop more effective preventive strategies and novel therapeutic approaches. By targeting the microbiota, interventions could be created to manage metabolic diseases, potentially shifting the landscape of NCD prevention and treatment. Some of these strategies could be applied at the primary care level, offering opportunities for early intervention and prevention from the first point of contact with healthcare systems.
1.- Bu F, Yao X, Lu Z, Yuan X, Chen C, Li L, Li Y, Jiang F, Zhu L, Shi G, Chen Y. Pathogenic or Therapeutic: The Mediating Role of Gut Microbiota in Non-Communicable Diseases. Front Cell Infect Microbiol. 2022 Jul 7;12:906349.

Reviewer 3 Report
Comments and Suggestions for Authors
Dear Editors of Biomedicines and Esteemed Authors of the manuscript titled “Primary prevention strategy for non-communicable diseases (NCDs) and their risk factors: the role of intestinal microbiota,” which has been submitted to the MDPI system, thank you for letting me review this work. NCDs increase morbidity and mortality, lead to uncontrollable health consequences, and significantly impact life quality. Nowadays, the control of NCDs is primarily based on lifestyle interventions for those who adhere to exercise and healthy eating. For those who do not, medications are helpful. These measurements encompass primary health care and secondary and tertiary health attention. The onset of NCDs based on gut microbiota alterations is relatively new, and there is the related novelty of this interesting work by Lopez-Tenorio et al. Before the Biomedicines Editor can decide, I would like to make some comments.
Major points
1. The flow of the abstract seems illogical in its final part. The authors seemed to attribute the gut microbiota alterations to unhealthy life habits. First, the authors cite NCDs as derived from bad life habits. Following this, the authors cite NCDs as partially derived from gut microbiota alterations. Finally, the authors mention that lifestyle interventions can significantly alter gut microbiota without context between the initial ideas. The authors must address this concern.
2. The authors must clearly state the novelty or the literature gap they want to cover with this review in the introduction section. The novelty or the literature gap they want to fill is completely missing.
3. There are areas of the manuscript that would benefit from further improvement. For example, in Lines 383-386 on Page 9, the authors gave their opinions about the content evaluated above. Paragraphs like this repeat along the manuscript, such as in Lines 535-537 on Page 12, Lines 553-556 on Page 13, etc. The author’s opinions regarding the issues discussed are highly warranted. The authors must try to increase the volume of content they opine, giving specific future research directions for research endeavors and insights into how the content discussed might influence patients attending primary health care or even specialized health attention. The authors must address these issues in all parts of the manuscript in which they provide opinions.
4. One of the most excellent lifestyle interventions is using functional foods to treat metabolic disorders. In Lines 396-414 on Page 10, the authors state the dietary interventions associated with gut microbiota alterations for the better. However, they do not state the current-of-art literature on functional food portions consumed daily to improve gut health. It is warranted since naturally occurring phytochemicals in functional foods play a role in treating gut alterations related to the intestinal microbiota.
5. The same must be done in stress management and sleep hygiene sections. The authors must try to address their point of view by adding specific recommendations from RCTs or even guidelines if they exist.
6. Probiotics, prebiotics, and synbiotics are explained in the manuscript. MAFLD is also a point of discussion. Probiotics, prebiotics, and synbiotics are great adjuvant treatment options for NAFLD-MAFLD/NASH today. There are significant papers delving into these aspects, such as 10.3390/ijms23158805 published by IJMS. Since this article received 67 citations in just two years, this represents the significance of the field. The authors must delve into the treatment of probiotics, prebiotics, and synbiotics more extensively in the manuscript. Of course, a dedicated section is not warranted, but at least the conclusions of the systematic review aforementioned must be discussed.
7. The role of systemic inflammation in metabolic diseases is well mentioned in the manuscript. However, we know today that meta-inflammation, a more global physiopathology mechanism, is well documented. The authors must state that in their manuscript and reformulate Section 3 to reflect the advancements in the field. For example, to highlight the importance of meta-inflammation, it is known that meta-inflammation targets metabolic reprogramming of immunological cells, such as macrophages, in treating diabetes and obesity. Do dietary or lifestyle interventions have a role to play in meta-inflammation reprogramming?
8. The authors must provide the limitations of their review, which must be clearly stated in the conclusion section.
Minor points
9. The reference style is not consistent with the MDPI’s style. The references are cited wrongly in the manuscript body.
10. You should remove the dot before the “Conclusions” heading in Section 8.
11. The introductory sentences in the manuscript body for the figures must address their main points, not only cited between parenthesis.
Thank you for your time.
Comments on the Quality of English LanguageMinor editing of English language required.
Author Response
Reviewer 3
Q1: The flow of the abstract seems illogical in its final part. The authors seemed to attribute the gut microbiota alterations to unhealthy life habits. First, the authors cite NCDs as derived from bad life habits. Following this, the authors cite NCDs as partially derived from gut microbiota alterations. Finally, the authors mention that lifestyle interventions can significantly alter gut microbiota without context between the initial ideas. The authors must address this concern.
A1: We thank the reviewer for the critical suggestion. In agreement, we have better described abstract section (please see the lines 50-57).
NCDs are highly prevalent worldwide, prompting increased attention to strategies for modifying the intestinal microbiota (IM). Approaches such as probiotics, prebiotics, synbiotics, and fecal transplantation (FMT) have demonstrated improvements in the quality of life for individuals with these conditions. Additionally, lifestyle changes and the adoption of healthy habits can significantly impact IM and may help prevent chronic diseases related to metabolism. Therefore, the main aim of this review is to analyze and understand the importance of microbiota intervention in the prevention of non-communicable diseases.
Q2: The authors must clearly state the novelty or the literature gap they want to cover with this review in the introduction section. The novelty or the literature gap they want to fill is completely missing.
A2: We thank the reviewer for the critical and appropriate suggestion. In agreement, we have better described the section (please see the lines 65-67).
NCDs, have a high mortality rate, registering 41 million deaths per year, that is, 74% of deaths worldwide. These diseases are characterized by remaining for a long period of time and evolving slowly. The search for alternatives to reduce these percentages of deaths leads us to propose knowledge alternatives and advance the study of them, as is the case of the Microbiota.
2.- Global Burden of Disease Collaborative Network, Global Burden of Disease Study 2019 (GBD 2019) Results (2020, Institute for Health Metrics and Evaluation – IHME)
Q3: There are areas of the manuscript that would benefit from further improvement. For example, in Lines 383-386 on Page 9, the authors gave their opinions about the content evaluated above. Paragraphs like this repeat along the manuscript, such as in Lines 535-537 on Page 12, Lines 553-556 on Page 13, etc. The author’s opinions regarding the issues discussed are highly warranted. The authors must try to increase the volume of content they opine, giving specific future research directions for research endeavors and insights into how the content discussed might influence patients attending primary health care or even specialized health attention. The authors must address these issues in all parts of the manuscript in which they provide opinions.
A3. We thank the reviewer for the critical suggestion. We hope to have satisfied the reviewer expectations.
Q4: One of the most excellent lifestyle interventions is using functional foods to treat metabolic disorders. In Lines 396-414 on Page 10, the authors state the dietary interventions associated with gut microbiota alterations for the better. However, they do not state the current-of-art literature on functional food portions consumed daily to improve gut health. It is warranted since naturally occurring phytochemicals in functional foods play a role in treating gut alterations related to the intestinal microbiota
A4: We thank the reviewer for the right suggestion
Recommendations are given on the consumption of foods with bioactive activity, since these consumption quantities are not stipulated for these foods that have a positive impact on health. The adjustments and recommendations that are suggested refer to consuming a variety of foods: they should be chosen from all food groups, including fruits, vegetables, whole grains, proteins and dairy products.
Vegetables can be consumed abundantly and fruits are preferably colorful since the variety of colors indicates the presence of phytochemicals, which are bioactive compounds that promote health, often found in colorful plant foods. (1)
Functional foods or foods with biological activity have the potential to treat digestive disorders mainly by playing a role in modulating inflammation. Reported studies suggest that compounds from some foods such as pomegranate, guava, onion, and red fruits promote an anti-inflammatory state, reducing the risk of generating chronic degenerative diseases, through the activation of certain signaling pathways, and also activating metabolism genes such as cholesterol 2. For this reason, it is emphasized that the consumption of these components than are closely related with immune response and the intestinal microbiota.
Essa MM, Bishir M, Bhat A, Chidambaram SB, Al-Balushi B, Hamdan H, Govindarajan N, Freidland RP, Qoronfleh MW. Functional foods and their impact on health. J Food Sci Technol. 2023 Mar;60(3):820-834.
López-Tenorio II, Domínguez-López A, Miliar-García Á, Escalona-Cardoso GN, Real-Sandoval SA, Gómez-Alcalá A, Jaramillo-Flores ME. Modulation of the mRNA of the Nlrp3 inflammasome by Morin and PUFAs in an obesity model induced by a high-fat diet. Food Res Int. 2020 Nov;137:109706
Q5: The same must be done in stress management and sleep hygiene sections. The authors must try to address their point of view by adding specific recommendations from RCTs or even guidelines if they exist.
A5: We thank the reviewer for the critical and appropriate suggestion
5.3 Stress Management
The World Health Organization implemented a guide where it shows the implementation of techniques for stress management: In times of stress, do what matters, an illustrated guide. Therefore, a review of it is suggested if the individual is going through a particular situation.
www.who.int/mental_health
5.4 Sleep Hygiene
To maintain adequate sleep hygiene, it is important for the individual to consider establishing rules in a constant and controlled manner to follow.
How to establish regular schedules, so you should stay active during the day so that you have at least 8 hours of rest at night, which is essential and necessary.
It is recommended to sleep in a way that generates comfort, avoid excessive consumption of breath before going home, since it could reconcile the floor more quickly in addition to not drinking alcohol, coffee, tea or chocolate. It is also recommended to avoid noises and lights that are annoying, such as the television or cell phone, and it is also suggested to avoid overthinking situations that cause us some type of concern. In addition to taking care of physical health. This brief guide is suggested www.nhlbi.nih.gov/sleep
Q6: Probiotics, prebiotics, and synbiotics are explained in the manuscript. MAFLD is also a point of discussion. Probiotics, prebiotics, and synbiotics are great adjuvant treatment options for NAFLD-MAFLD/NASH today. There are significant papers delving into these aspects, such as 10.3390/ijms23158805 published by IJMS. Since this article received 67 citations in just two years, this represents the significance of the field. The authors must delve into the treatment of probiotics, prebiotics, and synbiotics more extensively in the manuscript. Of course, a dedicated section is not warranted, but at least the conclusions of the systematic review aforementioned must be discussed.
A6: We thank the reviewer for the critical and appropriate suggestion. In agreement, we have better described Section 6.1 (please see the lines from 506 to 515).
Recent studies suggest that probiotics, prebiotics, and synbiotics are recognized for their effectiveness in treating MAFLD, including NAFLD and NASH. Guidelines emphasizing lifestyle changes—such as weight reduction, increased physical activity, and healthy diets—play a crucial role in managing NAFLD/NASH.
An imbalanced intestinal microbiome is associated with the pathogenesis of NAFLD/NASH. This GM dysbiosis can lead to alterations in the intestinal barrier and increased permeability, contributing to fat accumulation and, consequently, liver inflammation. This has directed therapy toward the intestinal microbiome as a novel therapeutic option. By modulating the intestinal environment, probiotics and synbiotics can reduce liver damage, improve metabolism, and decrease inflammation.
(, the lines 518 to 521).
Meta-analyses that address the effects of traditional probiotics in patients with NAFLD/NASH have demonstrated favorable therapeutic outcomes, including attenuation of inflammatory mediators, modulation of lipid metabolism, improvement of liver fibrosis, facilitation of weight control and obesity control.
(also the lines 536-538)
Synbiotics can further promote beneficial gut bacteria and the production of short-chain fatty acids (SCFAs), which have anti-inflammatory properties. This approach addresses some of the underlying factors that drive the progression of MAFLD and NASH.
(Finally the lines 543-546)
Recent studies report that probiotics/synbiotics can improve transaminase levels, hepatic steatosis, and NAFLD activity score. To some extent, probiotics/synbiotics can also reduce proinflammatory cytokines such as TNF-α and the interleukin family (IL-1, IL-6, IL-8).
1). Liu, J., & Goon, J. A. (2024). Roles of traditional and next-generation probiotics on non-alcoholic fatty liver disease (NAFLD) and non-alcoholic steatohepatitis (NASH): A systematic review and network meta-analysis. Antioxidants, 13(3), 329.
2). Xie, C., & Halegoua-DeMarzio, D. (2019). Role of probiotics in non-alcoholic fatty liver disease: Does gut microbiota matter? Nutrients, 11(11), 2837.
Q7: The role of systemic inflammation in metabolic diseases is well mentioned in the manuscript. However, we know today that meta-inflammation, a more global physiopathology mechanism, is well documented. The authors must state that in their manuscript and reformulate Section 3 to reflect the advancements in the field. For example, to highlight the importance of meta-inflammation, it is known that meta-inflammation targets metabolic reprogramming of immunological cells, such as macrophages, in treating diabetes and obesity. Do dietary or lifestyle interventions have a role to play in meta-inflammation reprogramming?
A7: We thank the reviewer for the critical and appropriate suggestion. In agreement, we have better described 3.0 section (please see the lines from 183 to 200).
Metainflammation is a relevantevent of chronic metabolic diseases, such as obesity and diabetes, that goes beyond systemic inflammation. It is characterized by chronic low-grade inflammation triggered by metabolic imbalance, leading to a reprogramming of immune cells, affecting macrophages in particular. These cells, which normally help maintain tissue homeostasis, undergo a shift from an anti-inflammatory (M2) to a pro-inflammatory (M1) phenotype during metainflammation. This change exacerbates insulin resistance and other metabolic disorders.
Macrophages play a crucial role in this process by altering their metabolic programming. In obesity and type 2 diabetes, they become dependent on glycolysis, which promotes the production of pro-inflammatory cytokines and contributes to insulin resistance and tissue damage. On the other hand, M2 macrophages, which depend on oxidative phosphorylation (OXPHOS) and fatty acid oxidation (FAO), are associated with anti-inflammatory effects. Reprogramming these macrophages back to an anti-inflammatory state is a key goal in the management of metabolic diseases.
Dietary and lifestyle interventions have been shown to influence this reprogramming. For example, calorie restriction, exercise, and anti-inflammatory diets can promote the shift of macrophages to the M2 phenotype. This reduces systemic inflammation and improves metabolic outcomes such as insulin sensitivity and lipid metabolism.
35.- Russo, S., Kwiatkowski, M., Govorukhina, N. y Melgert, B. N. (2021). Metainflamación y reprogramación metabólica de los macrófagos en la diabetes y la obesidad: la importancia de los metabolitos. Frontiers in Immunology, 12, 746151.
36.- Hu, T., Liu, CH., Lei, M. et al. Metabolic regulation of the immune system in health and diseases: mechanisms and interventions. Sig Transduct Target Ther 9, 268 (2024).
Q8: The authors must provide the limitations of their review, which must be clearly stated in the conclusion section.
A8: We thank the reviewer for the critical and appropriate suggestion. In agreement, we have better described conclusions section (please see the lines from 705 to 716).
There is a need for future research to better understand the GM role, as there are still relevant limitations to address. While the currently used adjuvant therapies, which involve specific bacterial colonies from healthy donors or supplements, have been shown to be safe in the short term, various risks may arise. These include issues related to donor availability and ensuring that the selected bacterial profile is appropriate for treating the specific non-communicable disease.
In cases where treatment is based on a supplement, it is essential that the product meets established standards and demonstrates proven effectiveness. This includes ensuring an appropriate abundance and diversity of species that can effectively repopulate the gut niche, which has shown positive health effects. Additionally, it is needed to consider that the costs of these therapies can sometimes be high., it is important to consider that the costs of these therapies can sometimes be high.
Pinto Y, Bhatt AS. Sequencing-based analysis of microbiomes. Nat Rev Genet. 2024 Jun 25.
Minor points
Q9: The reference style is not consistent with the MDPI’s style. The references are cited wrongly in the manuscript body.
A9: We corrected with MDPI’s style.
Q10: You should remove the dot before the “Conclusions” heading in Section 8.
A10: We have corrected this section.
Q11: The introductory sentences in the manuscript body for the figures must address their main points, not only cited between parenthesis.
A11: We thank the reviewer for the critical and appropriate suggestion. We made changes to the figure, and presented only one figure. Additionally, we describe the figure and add it in the figure caption and we have better described section Figure 1 (please see the lines 410-423) We hope to have satisfied the reviewer expectations.
Figure 1: Gut microbiota (GM) in eubiosis and dysbiosis and associated metabolites
- A) In a state of eubiosis, GM is characterized by a diverse and rich bacterial community that promotes the production of beneficial metabolites, like short-chain fatty acids (SCFAs). These metabolites have anti-inflammatory properties, regulate cholesterol and glucose metabolism and provide energy and maintain the integrity of the gut barrier, reducing the risk of degenerative diseases and preserving the gut barrier.
- B) The figure also illustrates the consequences of intestinal dysbiosis and loss of gut permeability, caused by an imbalance in microbial composition and bacterial translocation. This state is marked by an increase in levels of harmful metabolites, such as trimethylamine N-oxide (TMAO) and lipopolysaccharides (LPS), which contribute to increased intestinal permeability, endotoxemia, and chronic inflammation. Dysbiosis also activates Toll-like receptor 4 (TLR4), triggering an inflammatory cytokine cascade. Elevated TMAO and LPS levels have been linked to insulin resistance, inflammation, and a higher risk of metabolic disorders, including obesity, diabetes type 2 (T2DM), systemic arterial hypertension and cardiovascular diseases (CVD).
Reviewer 4 Report
Comments and Suggestions for Authors
The intestinal microbiota, consisting of microorganisms present in the intestine, decisively influences the health status of an individual. Variations in the composition and gene expression of the intestinal microbiota are associated with the risk of developing various diseases of the gastrointestinal tract (chronic inflammatory bowel diseases, irritable bowel syndrome, celiac disease, etc.), but also seem to be involved in the onset and progression of important systemic non-communicable diseases (allergies, metabolic diseases, oncological diseases, neurodegenerative and chronic neuro-inflammatory diseases, etc.). Current scientific evidence supports the idea that the composition and activity of the intestinal microbiota play a crucial role in influencing growth, development, aging and disease. This review aims to analyze and understand the relevance of the intestinal microbiota in the prevention of non-communicable diseases.
The topic covered in this review is very interesting and highly topical.
The Introduction section provides basic information on the research topic and outlines the importance of the study.
The following sections are well divided and provide an in-depth overview and analysis of the existing knowledge, research and scientific literature on the topic.
However, the authors could improve the quality of their work, add significant value to the current knowledge and build a solid foundation for future research by making some changes and additions, as reported below.
The authors should add a paragraph and references on recent findings with important health implications, which highlight that the human microbiome is highly transmissible. In fact, although research so far has confirmed and defined that the first transmission of the gut microbiome occurs at birth (vertical transmission), another decisive source of microbes that contribute to health are the people with whom we live in close contact (horizontal transmission). This research paves the way to understand how we receive microbial species associated with the risk of cardiovascular disease, diabetes, cancer and other pathologies.
The Conclusions section summarizes the level of knowledge existing between gut microbiota and possible primary prevention strategies for non-communicable diseases. Overall, it is well written, however, it should more strongly emphasize any remaining gaps and highlight areas requiring further research.
References should be cited in the text as required in the Instructions for Authors.
Add a list of the main acronyms used in the text.
Comments on the Quality of English LanguageThe review text uses easily readable English.
Author Response
Reviewer 4
The intestinal microbiota, consisting of microorganisms present in the intestine, decisively influences the health status of an individual. Variations in the composition and gene expression of the intestinal microbiota are associated with the risk of developing various diseases of the gastrointestinal tract (chronic inflammatory bowel diseases, irritable bowel syndrome, celiac disease, etc.), but also seem to be involved in the onset and progression of important systemic non-communicable diseases (allergies, metabolic diseases, oncological diseases, neurodegenerative and chronic neuro-inflammatory diseases, etc.). Current scientific evidence supports the idea that the composition and activity of the intestinal microbiota play a crucial role in influencing growth, development, aging and disease. This review aims to analyze and understand the relevance of the intestinal microbiota in the prevention of non-communicable diseases.
The topic covered in this review is very interesting and highly topical.
The Introduction section provides basic information on the research topic and outlines the importance of the study.
The following sections are well divided and provide an in-depth overview and analysis of the existing knowledge, research and scientific literature on the topic.
However, the authors could improve the quality of their work, add significant value to the current knowledge and build a solid foundation for future research by making some changes and additions, as reported below.
Q1. The authors should add a paragraph and references on recent findings with important health implications, which highlight that the human microbiome is highly transmissible. In fact, although research so far has confirmed and defined that the first transmission of the gut microbiome occurs at birth (vertical transmission), another decisive source of microbes that contribute to health are the people with whom we live in close contact (horizontal transmission). This research paves the way to understand how we receive microbial species associated with the risk of cardiovascular disease, diabetes, cancer and other pathologies.
A1: We thank the reviewer for the critical and appropriate suggestion. We hope to have satisfied the reviewer expectations.
This contribution helps us partially understand how certain microbial species are associated with the risk of presenting cardiovascular diseases, diabetes, cancer and other pathologies. It is important to highlight that the intestinal microbiota exhibits variations even among individuals from the same family, meaning each person has a unique microbiota profile that distinguishes them from others. Furthermore, the diversity and abundance of species within the intestinal microbiota can be modified by various factors, including race, gender, age, diet, lifestyle habits, exercise, and body composition. These factors can all influence an individual's microbiota footprint.
1.- Leite G, Pimentel M, Barlow GM, Chang C, Hosseini A, Wang J, Parodi G, Sedighi R, Rezaie A, Mathur R. Age and the aging process significantly alter the small bowel microbiom
Q2. The Conclusions section summarizes the level of knowledge existing between gut microbiota and possible primary prevention strategies for non-communicable diseases. Overall, it is well written, however, it should more strongly emphasize any remaining gaps and highlight areas requiring further research.
A2: We thank the reviewer for the constructive suggestion. In agreement, we have better described section conclusion (please see the lines 718-724). We hope to have satisfied the reviewer expectations.
In this field, there are multiple aspects that remain to be investigated due to deficiencies in both research and clinical practices that support personalized treatment. It is correct to recognize some limits in the comprehensive study of the intestinal microbiota, which opens up diverse possibilities for further exploration. This ongoing research can lead to new methodologies, diagnostic strategies, and treatment options for studying and understanding the microbiome. Additionally, it enhances the knowledge of healthcare personnel on the front lines of patient care.
Q3. References should be cited in the text as required in the Instructions for Authors.
A3. I appreciate your timely observation, which is why the references have been reviewed and modified according to the format requested by the magazine.
Q4. Add a list of the main acronyms used in the text.
A4: I would like to express my gratitude for this contribution, as I consider it essential. Therefore, a list of abbreviations has been compiled and added at the end of the manuscript page 37-38.
|
Bas |
Bile acids |
|
BMI |
Body Mass Index |
|
EPO FAO |
Erythropoietin Fatty acid oxidation |
|
FXR |
Farnesoid X Receptor |
|
GLP-1 and GLP-2 |
Glucagon-Like Peptide 1 and 2 |
|
GM |
Gut Microbiota |
|
HIF-1 |
Hypoxia-Inducible Factor 1 |
|
HOMA-IR |
Homeostatic Model a Assessment of Insulin Resistance |
|
IgA |
Immunoglobulin A |
|
LPS |
Lipopolysaccharide |
|
MAFLD |
Metabolic Dysfunction-Associated Fatty Liver Disease |
|
MS OXPHOS |
Metabolic Syndrome Oxidative phosphorylation |
|
NCDs |
Non-Communicable Diseases |
|
NF-kB |
Nuclear Factor kappa B |
|
PHC |
Primary Healthcare |
|
PPAR |
Proliferator-Activated Receptors |
|
SAH |
Systemic Arterial Hypertension |
|
T2D |
Type 2 Diabetes |
|
TLR4 |
Toll-like Receptor 4 |
|
TMA |
Trimethylamine |
|
TMAO
|
Trimethylamine N-oxide
|

Round 2
Reviewer 1 Report
Comments and Suggestions for Authors
The comments have been adequately addressed
Reviewer 2 Report
Comments and Suggestions for Authors
Thanks for your revisions.
Reviewer 3 Report
Comments and Suggestions for Authors
Esteemed Authors, thank you for addressing and considering my comments.
With best regards,
The Reviewer
Comments on the Quality of English LanguageMinor editing of English language required